# Synthesis of Microporous Mo_2_C-W_2_C Binary Carbides by Thermal Decomposition of Molybdenum-Tungsten Blues

**DOI:** 10.3390/nano10122428

**Published:** 2020-12-04

**Authors:** Natalia Gavrilova, Maria Myachina, Victor Dyakonov, Victor Nazarov, Valery Skudin

**Affiliations:** 1Department of Colloid Chemistry, Faculty of Natural Science, D. Mendeleev University of Chemical Technology of Russia, Miusskaya Sq., 9, 125047 Moscow, Russia; mmyachina@muctr.ru (M.M.); nazarov@muctr.ru (V.N.); 2JSC “Kompozit”, Pionerskaya Str. 4, Moscow Region, 141070 Korolev, Russia; v.dyakonov@mail.ru; 3Department of Chemical Technology of Carbon Materials, Faculty of Petroleum Chemistry and Polymers, D. Mendeleev University of Chemical Technology of Russia, Miusskaya Sq., 9, 125047 Moscow, Russia; skudin@muctr.ru

**Keywords:** binary carbides, molybdenum-tungsten carbide, sol-gel method, molybdenum-tungsten blues, thermal decomposition

## Abstract

Molybdenum and tungsten carbides are perspective catalytic systems. Their activity in many reactions is comparable to the activity of platinum group metals. The development of the synthesis method for of highly dispersed binary molybdenum and tungsten carbides is an important task. Dispersions of molybdenum-tungsten blue were used as a precursor for synthesis of binary molybdenum and tungsten carbides. The synthesis of carbides was carried out by thermal decomposition of molybdenum-tungsten blue xerogels in an inert atmosphere. The binary carbides were characterized by XRD, TGA, SEM and nitrogen adsorption. The influence of the molar ratio reducing agent/Me [R]/[ΣMe], molar ratio molybdenum/tungsten [Mo]/[W] on phase composition, and morphology and porous structure of binary carbides was investigated. Samples of binary molybdenum and tungsten carbides with a highly developed porous structure and a specific surface area were synthesized.

## 1. Introduction

Molybdenum and tungsten carbides are compounds with high electrical conductivity, thermal stability and mechanical strength [1]. Molybdenum and tungsten carbides possess a high catalytic activity in many catalytic processes. They are of great interest as catalysts for hydrogenation processes and natural gas conversion processes, as well as for use in fuel cells [2,3,4,5]. Activity of molybdenum and tungsten carbide is comparable to the activity of noble metals. Another advantage of carbides is their resistance to sintering and to many catalytic poisons [6,7,8].

Modern studies of the synthesis of these materials are carried out in two ways: improving the catalytic properties and increasing the dispersion of synthesized carbides [9,10,11,12]. An increase in catalytic activity can be achieved if a carbide of a metal similar in properties to molybdenum (for example, tungsten carbide [13,14]) is introduced into the crystal lattice of a highly active molybdenum carbide. Tungsten carbide is characterized by a higher oxophilicity, and when it is incorporated into the structure of molybdenum carbide, the metallic properties of molybdenum carbide change [15]. The presence of a synergistic effect in binary carbide Mo_2_C-W_2_C was confirmed for many catalytic processes including hydrogen evolution reaction (HER), dry reforming of methane, and conversion of aromatic hydrocarbons [10,16,17].

The two most important key factors in the strategy for the synthesis of a highly dispersed solid solution of molybdenum and tungsten carbides are at what stage will the formation of a binary compound of molybdenum and tungsten occurred and how will a high dispersion of the resulting material be achieved.

Traditional solid–solid methods for the preparation of molybdenum and tungsten carbides were also used for the synthesis of binary carbides [1,18]. However, due to high temperatures the products have incredibly low porosity and specific surface area for further catalytic application.

The most widely used method for synthesis of binary carbide is temperature-programmed reduction (TPR), this method related to gas–solid reactions. In the first stage, binary trioxides MoO_3_-WO_3_ are synthesized, which are then subjected to temperature-programmed carburization in the second stage. The synthesis of trioxides can be carried out by different methods: mechanical stirring, coprecipitation, hydrothermal synthesis, and freeze-drying [10,19,20]; electrochemical deposition [21]; chemical vapor deposition [17]; precipitation from solutions [22]; thermal decomposition of salts [23]; or the sol-gel method [24]. Gas mixture of methane/hydrogen or propane/hydrogen [25,26,27,28,29] as a carburizing agent are usually used.

The most promising group of methods is based on liquid–liquid reactions, which make it possible to synthesize carbides in two stages under mild conditions. This group is united by the idea of using water-soluble organic substances as a carbon source. In this case, the formation of carbide occurs at the stage of heat treatment of molybdenum or tungsten compounds and an organic compound (glucose, urea, hexamethylenetetramine etc.). This approach makes it possible to combine the stage of heat treatment and the stage of carbide formation.

In [30,31], molybdenum carbide was synthesized using urea and molybdenum pentachloride. Mixing reagents with ethyl alcohol resulted in formation of gel, the thermal treatment of which in an inert medium led to the formation of Mo_2_C with specific surface area of 22 m^2^/g. In [32], gel formation happened after adding ammonium heptamolybdate to the sucrose solution. It was shown that the formation of molybdenum carbides occured after drying and heating gel in inert atmosphere. 

A similar synthesis was carried out to obtain tungsten carbide [33]. The starting reagents were ammonium paratungstate, nitric acid, urea, and glucose. When mixing the reagents, a gel was formed, which was then dried, and heat treated in an Ar flow at various temperatures in the range of 800–1100 °C. Under certain conditions, WC was formed with a particle size of less than 200 nm. The phase composition of the synthesis products was determined by the [W]/[glucose] ratio. 

The synthesis of molybdenum and tungsten carbides using solutions and a carbon-containing precursor (organic substances) is the most suitable route, since it allows varying the composition of particles over a wide range, as well as carrying out the process at temperatures (800–900 °C) without use of explosive gas mixtures. An important advantage of using organic substances is the ability to regulate the [Me]/[organic] ratio that avoids the formation of free carbon on the surface of the particles.

Besides solutions of molybdates as a source of molybdenum, dispersions of polyoxometalate clusters (POM) can be used. In [34,35,36,37], preparation of molybdenum carbide by thermal decomposition of molybdenum blue, synthesized with different organic reducing agent, was described. Phase composition, porosity, and surface area is determined by the type and amount of reducing agent used in the synthesis of molybdenum blue. 

Molybdenum blues are dispersions of molybdenum oxide clusters up to eight nm in size [38]. These compounds can be obtained by reduction of molybdates in an acidic medium. Particle formation occurs during the self-organization of the original building blocks. Depending on the synthesis conditions, the formation of a series of giant molybdenum oxide clusters of various compositions (Mo_134_, Mo_142_, Mo_158,_ etc.) occurs [39,40]. 

It is known that under certain conditions, tungsten also forms tungsten blues containing tungsten oxide nanoclusters [41]. Carrying out the reduction reaction in a mixture of salts, the formation of molybdenum-tungsten blue is possible, which can be used as precursors for the synthesis of binary carbides.

The aim of this work was to synthesize a highly dispersed binary molybdenum and tungsten carbides using molybdenum-tungsten blue and an organic reducing agent.

## 2. Materials and Methods 

### 2.1. Synthesis of Molybdenum-Tungsten Blue Dispersion

For the synthesis of molybdenum-tungsten blue, ammonium heptamolybdate (NH_4_)_6_Mo_7_O_24_·4H_2_O, ammonium tungstate (NH_4_)_10_W_12_O_41_·5H_2_O, ascorbic acid C_6_H_8_O_6_, hydrochloric acid HCl were used. All chemicals were purchased in CT Lantan (Moscow, Russia) and used without further purification. 

Dispersions of molybdenum-tungsten were synthesized by chemical reduction of acidified mixed solution of ammonium heptamolybdate and tungstate. Ascorbic acid was used as organic reducing agent, hydrochloric acid as an acidifying agent. 

The initial concentration of molybdate was 0.2 M, the initial concentration of tungstate was 0.1 M. The synthesis was carried out at the constant metal concentration 0.07 M (the sum of molybdenum and tungsten), at different molar ratios [R]/[ΣMe] = 0.8–5 and [Mo]/[W] = 50/50–100/0 and constant molar ratio [H]/[Me] = 0.6. The mixing of reagents was carried out during 1 h at the room temperature. The formation of molybdenum-tungsten nanoparticles occured within 1 day according to the dynamic light scattering results. The formation of molybdenum-tungsten blue dispersions was very similar to the formation of molybdenum blue dispersions [42].

### 2.2. Characterization of Molybdenum-Tungsten Blue Dispersion

Analysis of dispersions by dynamic light scattering (DLS) was performed using a Photocor Compact-Z instrument (Photocor LLC, Moscow, Russia) at a wavelength of 658 nm. The signal was accumulating during 10 min at a laser power of 25 mW. Micrographs of molybdenum-tungsten blue nanoparticles were obtained using a transmission electron microscope LEO 912AB Omega (Carl Zeiss, SMT AG Oberckochen, Germany).

### 2.3. Synthesis of Binary Carbides

Xerogels of molybdenum-tungsten blues were obtained by drying dispersions at room temperature. The synthesis of molybdenum and tungsten carbides was carried out by thermal decomposition of xerogels in an inert atmosphere (N_2_) at 900 °C in a tubular furnace with holding for 1 h at a given temperature. The temperature rise rate was 5 °C/min.

### 2.4. Binary Carbide Characterization

The molybdenum-tungsten blue xerogels were investigated by thermal analysis using an SDT Q600 thermal analyzer (TA Instruments, New Castle, DE, USA). Xerogel samples (10 mg) were calcined in inert atmosphere (Ar) in the temperature range from 25 to 1000 °C at a heating rate of 5 °C/min.

The phase composition of the samples was determined by X-ray diffraction (XRD) patterns using a Rigaku D/MAX 2500 X-ray diffractometer (Rigaku Corporation, Tokyo, Japan) with CuKα radiation. Patterns were recording at the scanning range (2θ) 20–110° with a step 0.02° and at scan rate 5°min^−1^. The phases were identified according to the JCPDS Powder Diffraction File data.

The morphology of the synthesized samples was investigated using a CAMSCAN S2 scanning microscope (Cambridge Instruments, Cambridge, UK) with accelerating voltage 20 kV. 

The porous structure of the samples was analyzed using the low-temperature isotherms of nitrogen adsorption. The studies were carried out on a Gemini VII (Micromeritics, Norcross, GA, USA) surface area and porosity analyzer at Mendeleev University Center for Shared Use of Equipment.

The specific surface area was determined by the Brunauer–Emmett–Teller method (BET). The total specific pore volume was calculated at the maximum value of the relative pressure equal to 0.995. The mesopore volume, the mesopore size distribution, and the predominant pore diameter were calculated by the Barrett, Joyner, and Halenda method (BJH). The micropore volume was determined using t-plot de Boer method, the thickness of the adsorption film was calculated using the Harkins-Jura equation. The predominant micropore size was calculated using the Horvath-Kawazoe method.

## 3. Results

### 3.1. Molybdenum-Tungsten Blue Dispersion Properties

For the synthesis of highly dispersed binary molybdenum and tungsten carbides, dispersions of molybdenum-tungsten blue were used. These dispersions are aqueous systems of molybdenum and tungsten polyoxometalate complexes. A specific property of these complexes was their big size (from 1 to 8 nm), which makes it possible to classify aqueous systems based on them as colloidal systems (dispersions or sols). 

Figure 1 shows TEM images and DLS distribution of molybdenum-tungsten blue particles. According to TEM data, all molybdenum-tungsten blue nanoparticles with different molar ratio [Mo]/[W] were characterized by a particle size of about 3–4 nm, which was confirmed by the DLS results, according to which the most probable hydrodynamic radius of particles was 1.5 nm. The results obtained agree with the data on the size of polyoxometalate complexes given in the literature [40]. 

It should be noted that one maximum was observed in the particle size distribution (Figure 1a–c), which indicated the predominance of one form of particles, the exception is the sample with the molar ratio [Mo]/[W] = 50/50. This was probably due to the presence of two different cluster shapes, differing in size.

The synthesized dispersions had chemical and aggregate stability in the pH range from 1.0 to 2.0. In the selected pH range, dispersions existed for a long time (more than 6 months) and no coagulation or sedimentation were occurring.

The formation of molybdenum-tungsten blue particles occured because of the reduction of molybdate and tungstate ions in an acidic medium. The choice of organic compounds (ascorbic acid) as a reducing agent made it possible to obtain stable dispersions of molybdenum-tungsten blue. The choice of the content of the reducing agent (molar ratio [R]/[ΣMe]) should be determined not only by the possibility of obtaining stable dispersions, but also by the possibility of obtaining binary carbides during the thermal treatment of xerogels in an inert medium.

The synthesis method developed in [42] allowed the synthesis of molybdenum blue dispersion with ascorbic acid in a wide range of molar ratios [R]/[Mo] from 0.8 to 5. Under similar conditions molybdenum-tungsten blues can also be obtained. However, it should be noted that molybdenum-tungsten blues dispersions with a molar ratio [R]/[ΣMe] less than 1 did not have sufficient aggregate stability. Therefore, further we considered dispersions synthesized in the [R]/[ΣMe] range from 1 to 5.

### 3.2. Synthesis of Binary Carbides

To determine the temperature of carbide formation, the differential thermogravimetry was used. DTG, DTA, and TG curves for samples of xerogels with different molar ratio [Mo]/[W] and constant molar ratio [R]/[ΣMe] = 5.0 are presented in Figure 2. In the temperature range of 50–200 °C for all molar ratios [Mo]/[W], there was a change in the sample mass associated with evaporating water in the interparticle space. When the temperature reached 200 °C, bound water presented in the molybdenum-tungsten blue in the form of a hydration shell left. The thermal effect of this process was negative, as can be seen from the given patterns.

With increasing temperature, thermolysis processes began to occur. In the temperature range 350–400 °C, the process of decomposition of ammonium chloride, which was presented in the blue molybdenum-tungsten blue dispersion medium, proceeds intensively. Thermolysis of ascorbic acid and its oxidation products was most intensive at the temperature of 700–750 °C.

In the temperature range 700–800 °C, there was a narrow peak in the DTA and DTG curves with a negative thermal effect. The presence of such a peak was characteristic of carbide phase formation.

It should be noted that an increase in the tungsten content led to an increase in the temperature of carbide formation from 700 °C for the system [Mo]/[W] = 95/5 to 800 °C for the system—[Mo]/[W] = 50/50, which was associated with higher temperature of formation of tungsten carbide.

To establish transformations of molybdenum-tungsten blue xerogels to binary carbides, XRD analysis was used. Figure 3 shows the XRD patterns for samples of molybdenum-tungsten blue xerogels with different molar ratio [Mo]/[W] calcined in the temperature range of 500–900 °C. As can be seen from the presented patterns, the formation of carbides occurs in the temperature range 700–900 °C. For the samples with low content of tungsten ([Mo]/[W] = 95/5; 90/10; 80/20), the oxides of molybdenum and tungsten were not observed. The formation of binary carbide Mo_2_C-W_2_C occured from amorphous precursor at 700 °C, which was comparable with values, presented in the literature [14].

Besides formation of binary carbide Mo_2_C-W_2_C, another phase—molybdenum carbide MoC—was also forming in samples. The presence of this carbide is typical for sol-gel method using molybdenum blue and ascorbic acid as a carbon source [34].

The phase transformation in the sample with molar ratio [Mo]/[W] = 50/50 was going by another way. First, molybdenum and tungsten dioxides were forming at 600 °C. Then these oxides were transforming to carbide phase, which included Mo_2_C-W_2_C, WC, and Mo.

Comparing the results of carbide formation from samples with different molar ratio [Mo]/[W], we made a conclusion, that in the case of [Mo]/[W] = 95/5; 90/10; 80/20, the formation of solid solution was possible due to the precursor polyoxometalate complexes, containing molybdenum and tungsten. The sample with molar ratio [Mo]/[W] = 50/50 was a mixture of molybdenum and tungsten carbides, which was assumed in the discussion of the DLS results (Figure 1d).

The content of the reducing agent in the dispersion of molybdenum-tungsten had a strong effect on the formation of molybdenum and tungsten carbides, mainly on the final carbon content in the resulting carbides. To establish the optimal molar ratio [R]/[ΣMe], XRD patterns were obtained. For example, XRD patterns for samples [Mo]/[W] = 50/50 calcined at 900 °C are shown in Figure 4.

As can be seen with an increase in the content of ascorbic acid, the intensity of reflections decreased, which corresponded to a decrease in the crystallinity of molybdenum and tungsten carbides. For all the molar ratios, the formation of molybdenum and tungsten carbide mixture was observed. At ratios less than 2.0, metal Mo was also forming besides molybdenum and tungsten carbide. At ratios more than 1.0, the formation of free carbon was observed. Molar ratio [R]/[ΣMe] = 1.0 was chosen for the synthesis of binary carbides as it makes possible to obtain carbides without an excess of free carbon.

The results discussed above were obtained for the equimolar ratio of molybdenum and tungsten [Mo]/[W] = 50/50. After establishing the conditions for the formation of carbides (900 °C, [R]/[Me] = 1.0), the next step was to establish the dependences of the formation of molybdenum and tungsten carbides with different tungsten contents and to confirm the formation of binary carbides. Samples with a molar ratio [Mo]/[W] = 95/5, 90/10, 80/20 were synthesized. For comparison, the results of the Mo_2_C obtained from molybdenum blue xerogels, synthesized with ascorbic acid in similar conditions, are shown.

Figure 5 shows XRD patterns of samples of molybdenum and tungsten carbides, obtained at a constant molar ratio [R]/[ΣMe] = 1/1. The introduction of small amounts of tungsten into the system with molybdenum carbide ([Mo]/[W] = 95/5, 90/10, 80/20) led to a change in the phase composition of the resulting materials in comparison with pure molybdenum carbide. Broad peaks correspond to solid solution formation and small crystalline size.

At the same time, the transition to the equimolar ratio of molybdenum and tungsten ([Mo]/[W] = 50/50) led to a sharp decrease in the intensity of the main reflections. In the case of the sample [Mo]/[W] = 50/50, in addition to the formation of tungsten carbide W_2_C, the formation of the WC occured. These compounds were observed for all samples obtained with the ratio [Mo]/[W] = 50/50 (see Figure 3d).

The morphology of binary carbides was studied by scanning electron microscopy (SEM). Figure 6 shows micrographs of the samples. Binary carbides were aggregates of primary particles; the size of primary particles was about 100 nm.

The porous structure of synthesized carbides was investigated by low-temperature nitrogen adsorption/desorption isotherms. Figure 7 shows adsorption isotherms for binary molybdenum and tungsten carbides synthesized with molar ratio [R]/[ΣMe] = 1. All isotherms can be classified as type IV according to Brunauer’s classification, and the hysteresis loop as type H4, which is characteristic of a material with slit pores geometry.

Isotherms for binary molybdenum-tungsten carbides have a region of high increase of adsorption at low pressure, which is representative for micropores. The presence of a microporous structure was also confirmed by the form of the t-plot (Figure 7b). Straight lines (Figure 7b) were tangent to the -plot. The cutoff on the ordinate axis allowed to determine the volume of micropores in the sample. It follows from the t-plot that the samples of molybdenum and tungsten carbides had both a mesoporous structure due to the interparticle space and a microporous structure arising from small particles size of binary molybdenum and tungsten carbides.

Meso- and micropore size distribution for synthesized binary carbides are shown in Figure 8. As can be seen, the predominant mesopore size for all samples was 4 nm and samples with molar ratio [Mo]/[W] = 95/5; 90/10; 80/20 had also second predominant micropore size—1.2 nm.

Table 1 shows the values of the specific surface area, pore volume and predominant pore size for molybdenum and tungsten carbides with different molar ratios [Mo]/[W].

For samples [Mo]/[W] = 95/5, 90/10 and 80/20, the BET specific surface area was almost 50 times higher than for pure molybdenum carbide obtained in the same conditions. The low specific surface area of the sample of molybdenum and tungsten carbides [Mo]/[W] = 50/50 was due to the fact that the formation of a solid solution did not occur and the dispersity of material was low. This sample is a mixture of molybdenum and tungsten carbides, which were characterized by a mesoporous structure.

A porous structure, including microporous and mesoporous, was characterized by the presence of the two most probable pore sizes of 4 nm (mesopore size) and 1.2 nm (micropore size) (see Figure 8). The fraction of micropore volume is much larger than the fraction of mesopores in the total pore volume. It is explained by the high value of the specific surface area of the solid solution of molybdenum and tungsten carbides.

Such changes in the systems [Mo]/[W] = 95/5, 90/10 and 80/20 can be explained by the incorporation of tungsten into the crystal lattice of molybdenum carbide, the formation of solid solutions with high dispersity references, and the phenomenon of broadening of diffraction lines associated with a decrease in the particle size [27,29]. For the first time, the possibility of the formation of solid solutions in the system was demonstrated in [43]. Then this was confirmed by researchers using different methods for the synthesis of carbides (metallurgical, electrochemical, chemical vapor deposition with further carburization) [17,21,44].

## 4. Discussion

Molybdenum and tungsten carbides were obtained by thermal decomposition of molybdenum-tungsten blue xerogels. The formation of a carbides in one stage was possible by the presence of a reducing agent, ascorbic acid, as a carbon source in dispersions.

According to the data of thermogravimetric analysis, it was found that the formation of molybdenum and tungsten carbides in an inert atmosphere occurs in the temperature range of 700–800 °C, depending on the tungsten content. An increase in the tungsten content led to an increase in the reaction temperature with the formation of carbides.

In samples with molar ratio [Mo]/[W] = 95/5, [Mo]/[W] = 90/10, [Mo]/[W] = 80/20, solid solution Mo_2_C-W_2_C was formed, while in the case of a sample with an equimolar ratio of molybdenum and tungsten, a mixture of molybdenum and tungsten carbides Mo_2_C, W_2_C, and WC was formed. Molar ratio[R]/[ΣMe] = 1/1 was chosen as the optimal molar ratio of the reducing agent to the metal, since it was precisely this content of the reducing agent in the precursor that made it possible to obtain carbides without additional amorphous carbon on the particle surface.

Analysis of the porous characteristics of the samples showed that the Mo_2_C-W_2_C systems ([Mo]/[W] = 95/5, [Mo]/[W] = 90/10, [Mo]/[W] = 80/20) contain micropores and mesopores with a narrow pore size distribution, and the sample [Mo]/[W] = 50/50, which consisted of carbide mixture Mo_2_C, W_2_C, and WC contained only mesopores.

It should be noted that methods for synthesis of solid carbide solutions and especially microporous solid solutions can be rarely found in literature [14,25]. Sol-gel method based on using molybdenum-tungsten blue dispersion made it possible to control phase composition and porous structure of molybdenum-tungsten carbides and synthesize microporous material with high surface area.

## 5. Conclusions

The strategy for the synthesis of molybdenum and tungsten carbides using the initial highly dispersed precursors, molybdenum-tungsten blue, allowed to obtain a microporous material with molar ratios [Mo]/[W] = 95/5; 90/10; 80/20. The synthesis method made it possible to obtain binary carbide without using a carburization step. It was found that carbides were formed in the range of 700–800 °C. Molar ratio [R]/[ΣMe] was a parameter that had a significant role in the formation and properties of binary carbide. The synthesized solid solutions of molybdenum and tungsten carbides had a developed microporous structure and a high specific surface area of about 150 m^2^/g.

The proposed method for the preparation of molybdenum and tungsten carbides was characterized by its simplicity, a comparatively low temperature of formation of the carbide, and allowed to synthesize a material with a developed porous structure.

## Figures and Tables

**Figure 1 nanomaterials-10-02428-f001:**
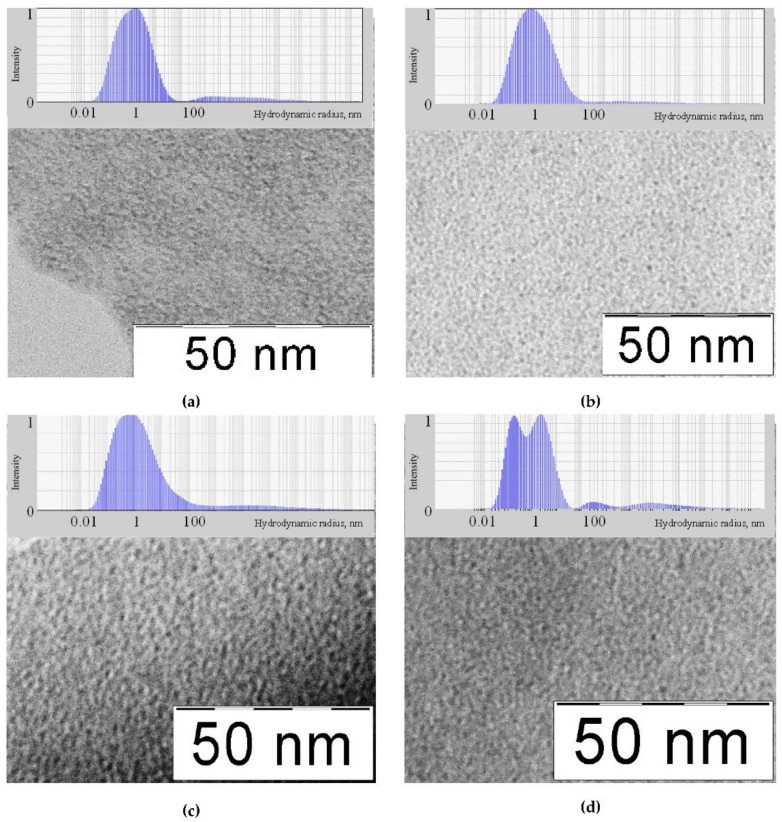
TEM image and dynamic light scattering (DLS) particle size distribution of molybdenum-tungsten blues with different molar ratio: (**a**) [Mo]/[W] = 95/5, (**b**) [Mo]/[W] = 90/10, (**c**) [Mo]/[W] = 80/20, (**d**) [Mo]/[W] = 50/50.

**Figure 2 nanomaterials-10-02428-f002:**
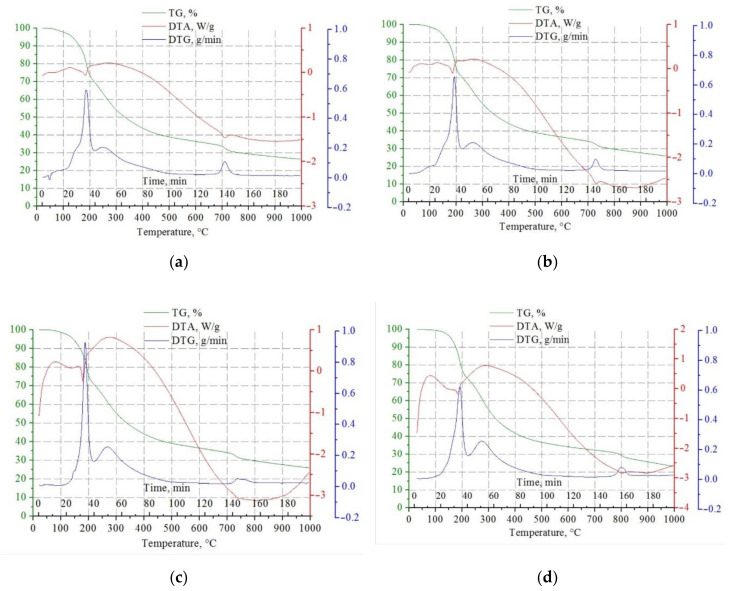
DTA curves of molybdenum-tungsten blue xerogel synthesized with: (**a**) [Mo]/[W] = 95/5; (**b**) [Mo]/[W] = 90/10; (**c**) [Mo]/[W] = 80/20; (**d**) [Mo]/[W] = 50/50. (molar ratio [R]/[ΣMe] = 5.0, inert atmosphere).

**Figure 3 nanomaterials-10-02428-f003:**
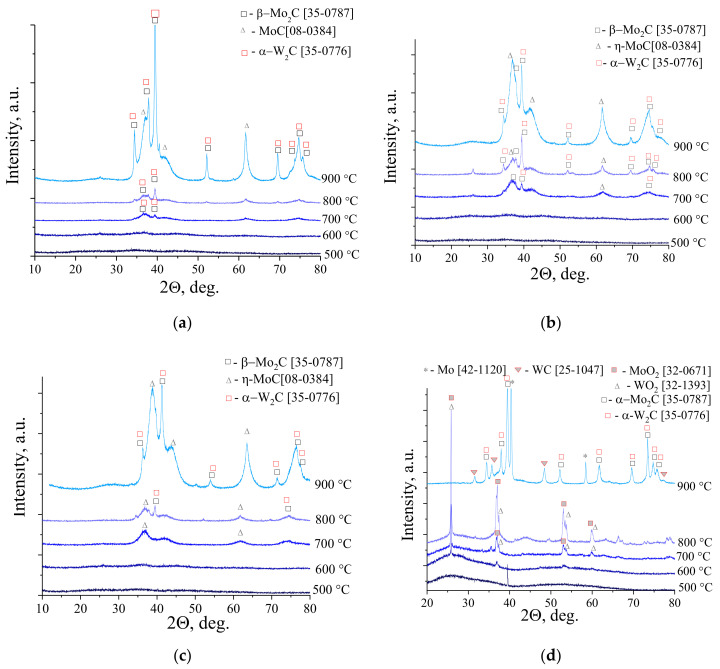
XRD patterns of molybdenum-tungsten blue xerogels collected in the function of temperature for the samples calcined at maximum 900 °C in inert atmosphere with different ratios: (**a**) [Mo]/[W] = 95/5; (**b**) [Mo]/[W] = 90/10; (**c**) [Mo]/[W] = 80/20; (**d**) [Mo]/[W] = 50/50 ([R]/[ΣMe] = 1).

**Figure 4 nanomaterials-10-02428-f004:**
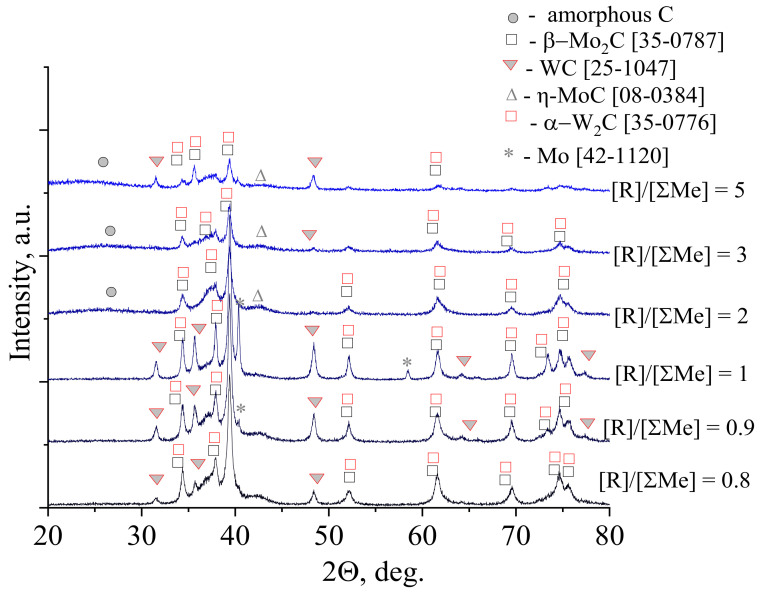
XRD pattern of molybdenum-tungsten carbides with molar ratio [Mo]/[W] = 50/50 calcined at 900 °C in inert atmosphere.

**Figure 5 nanomaterials-10-02428-f005:**
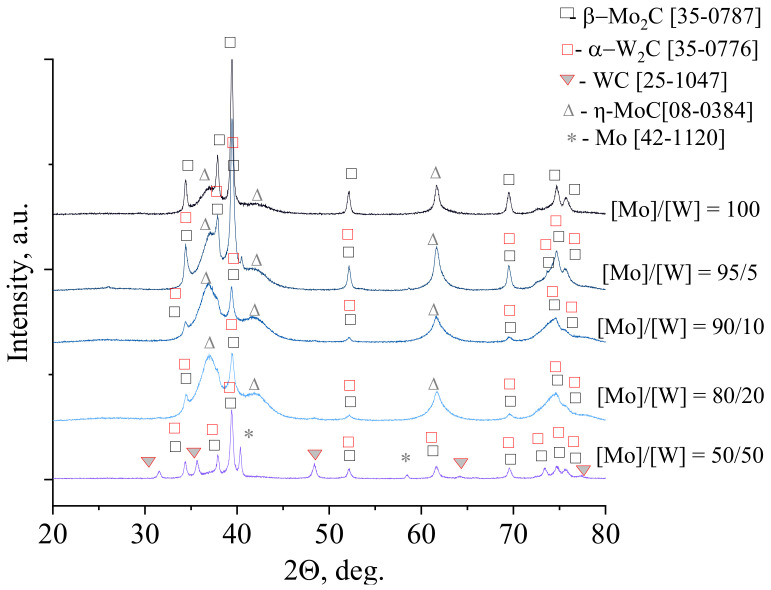
XRD pattern of molybdenum-tungsten carbides with molar ratio [R]/[ΣMe] = 1 calcined at 900 °C in inert atmosphere.

**Figure 6 nanomaterials-10-02428-f006:**
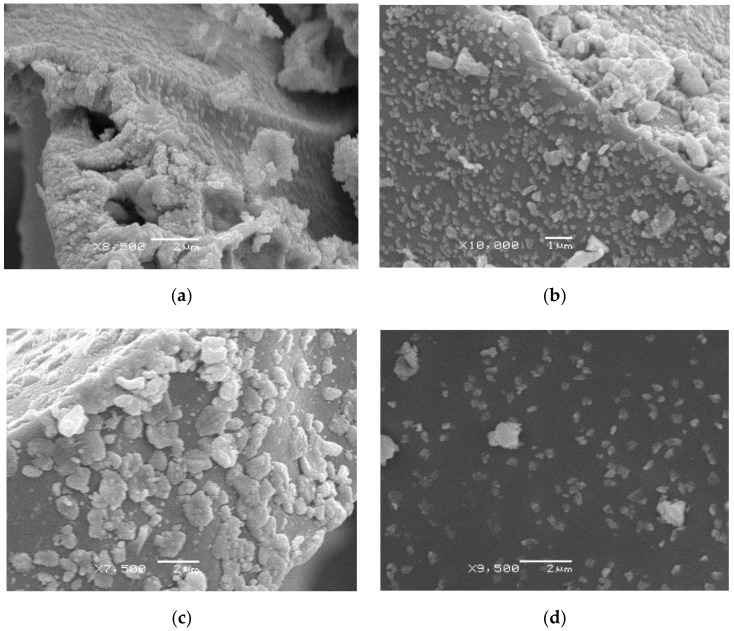
SEM-images on molybdenum-tungsten carbides synthesized from molybdenum-tungsten blue xerogels (calcined at 900 °C in inert atmosphere) with different ratios: (**a**) [Mo]/[W] = 95/5; (**b**) [Mo]/[W] = 90:10; (**c**) [Mo]/[W] = 80:20; (**d**) [Mo]/[W] = 50:50 and constant molar ratio [R]/[ΣMe] = 1).

**Figure 7 nanomaterials-10-02428-f007:**
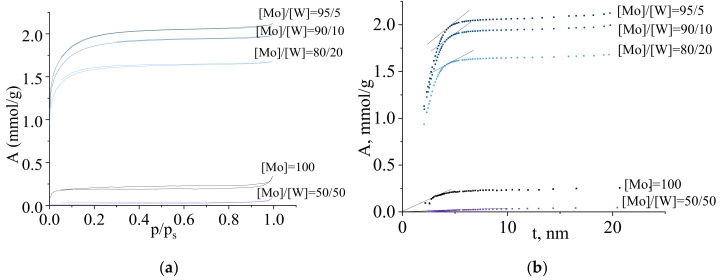
(**a**) Nitrogen adsorption/desorption isotherms and (**b**) t-plots on molybdenum-tungsten carbides synthesized from molybdenum-tungsten blue xerogels (calcined at 900 °C in inert atmosphere) with different ratios [Mo]/[W].

**Figure 8 nanomaterials-10-02428-f008:**
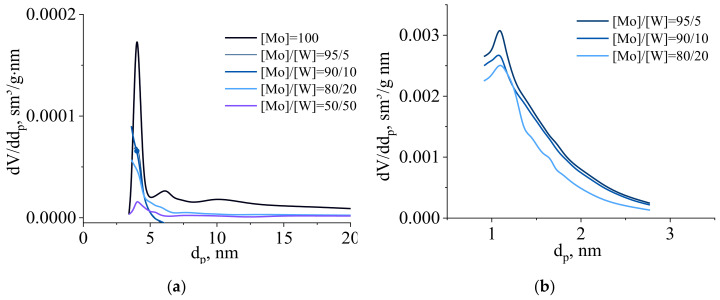
(**a**) Meso- and (**b**) micropore distribution of binary carbides, synthesized molybdenum-tungsten blue xerogels (calcined at 900 °C in inert atmosphere) with different ratios [Mo]/[W] and constant molar ratio [R]/[ΣMe] = 1).

**Table 1 nanomaterials-10-02428-t001:** Porous structure of molybdenum-tungsten carbides calcined at 900 °C in inert atmosphere ([R]/[Me] = 1).

Molar Ratio [Mo]/[W]	Parameters
Surface Area (BET), m^2^/g	Total Pore Volume, cm^3^/g	Mesopore Volume, cm^3^/g (BJH Desorption)	Micropore Volume, cm^3^/g (t-Plot)	Mesopore Diameter, nm (BJH)	Micropore Diameter, nm (Horvath-Kawazoe)
100	3.2	0.010	0.006	0.002	4	
95/5	150	0.073	0.005	0.68	4	1.2
90/10	141	0.068	0.002	0.064	4	1.2
80/20	120	0.058	0.002	0.055	4	1.2
50/50	1.4	0.002	0.002		4

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
