# Peer review of "Synthesis of Microporous Mo2C-W2C Binary Carbides by Thermal Decomposition of Molybdenum-Tungsten Blues"

_nanomaterials, 2020, doi:10.3390/nano10122428_

Round 1
Reviewer 1 Report
The manuscript by Gavrilova et al. presents the synthesis and characterization of molybdenum and tungsten carbides by thermolysis of molybdenum-tungsten blue xerogels in presence of ascorbic acid as a reducing agent and carbon source. The manuscript is clearly written and logically organized. The content is technically sound, reflecting the practices in this area of study. I have only minor points of criticisms. The description of the synthesis of molybdenum-tungsten blue dispersion (section 2.1, lines 98-107) should be presented in more detail. In particular, the concentrations of all the solutions used in the synthesis as well as the time of the reaction should be explicitly stated. The word "destruction" in line 187 should be replaced with "decomposition" (I believe that authors had in mind the thermal decomposition of ammonium chloride to ammonia and hydrochloric acid). In the Figure 3 caption the phrase "XRD pattern of molybdenum-tungsten blue xerogels calcined at 900 °C..." should be replaced with "XRD patterns of molybdenum-tungsten blue xerogels collected in the function of temperature for the samples calcined at maximum 900 °C...". The sentence in lines 238-239 is not clear; should it rather read as: "For comparison, the results of the Mo2C obtained from molybdenum blue xerogels, synthesized with ascorbic acid in similar conditions, are shown". Line 262: rewrite "t-plot (Fig. 6,b)" with "t-plot (Fig. 7b)". Besides, the meaning of the solid black lines in Fig. 7b, tangent to the experimental points shown as scatter plots, should be explained in the Figure 7 caption. The supposition of the incorporation of tungsten atoms in the crystal lattice of molybdenum carbide suggested in line 284, and in general the coupled substitution of Mo by W, should be supported by a comparison with literature data on similar systems. Finally, "phase transition temperature" (line 297) should be replaced with "reaction temperature".
Author Response
1. The manuscript by Gavrilova et al. presents the synthesis and characterization of molybdenum and tungsten carbides by thermolysis of molybdenum-tungsten blue xerogels in presence of ascorbic acid as a reducing agent and carbon source. The manuscript is clearly written and logically organized. The content is technically sound, reflecting the practices in this area of study.
1. The authors are grateful to the Reviewer for such a careful study of the manuscript.
2. I have only minor points of criticisms. The description of the synthesis of molybdenum-tungsten blue dispersion (section 2.1, lines 98-107) should be presented in more detail. In particular, the concentrations of all the solutions used in the synthesis as well as the time of the reaction should be explicitly stated.
2. The authors agree with the remark. The synthesis of molybdenum-tungsten blue is supplemented by the conditions for its implementation.
3. The word "destruction" in line 187 should be replaced with "decomposition" (I believe that authors had in mind the thermal decomposition of ammonium chloride to ammonia and hydrochloric acid).
3. The authors agree with the remark. Сorrections made to the text of the manuscript
4. In the Figure 3 caption the phrase "XRD pattern of molybdenum-tungsten blue xerogels calcined at 900 °C..." should be replaced with "XRD patterns of molybdenum-tungsten blue xerogels collected in the function of temperature for the samples calcined at maximum 900 °C...".
4. Сorrections made to the manuscript
5. The sentence in lines 238-239 is not clear; should it rather read as: "For comparison, the results of the Mo2C obtained from molybdenum blue xerogels, synthesized with ascorbic acid in similar conditions, are shown".
5. The authors agree with the remark. Сorrections made to the manuscript
6. Line 262: rewrite "t-plot (Fig. 6,b)" with "t-plot (Fig. 7b)". Besides, the meaning of the solid black lines in Fig. 7b, tangent to the experimental points shown as scatter plots, should be explained in the Figure 7 caption.
6. Сorrections made to the manuscript
7. The supposition of the incorporation of tungsten atoms in the crystal lattice of molybdenum carbide suggested in line 284, and in general the coupled substitution of Mo by W, should be supported by a comparison with literature data on similar systems.
7. Сorrections made to the manuscript.Comparison with literature data is given.
8. Finally, "phase transition temperature" (line 297) should be replaced with "reaction temperature".
8. The authors agree with the remark. Сorrections made to the manuscript
Reviewer 2 Report
The manuscript entitled "Synthesis of Microporous Mo2C-W2C Binary Carbides by Thermal Decomposition of Molybdenum-Tungsten Blues" was reviewed.
In my opinion, this manuscript is a valuable report on the synthesis of Mo2C-W2C g and technically sound. However, the “Discussion” section is not really discussion only contains a description of the results without justification or explanation of the reasons for the results. This report is not adequate for scientific writing and does not hold any scientific novelty and significance. Unfortunately, I could not give a positive comment in the current paper. I recommend that the manuscript is published as a Technical note, but not as a research paper.
Author Response
1. In my opinion, this manuscript is a valuable report on the synthesis of Mo2C-W2C g and technically sound. However, the “Discussion” section is not really discussion only contains a description of the results without justification or explanation of the reasons for the results. This report is not adequate for scientific writing and does not hold any scientific novelty and significance. Unfortunately, I could not give a positive comment in the current paper. I recommend that the manuscript is published as a Technical note, but not as a research paper.
1. The authors are grateful to the Reviewer for such a careful study of the manuscript. The authors cannot fully agree with the reviewer. Since the manuscript presents a new method for the synthesis of binary carbides. The material is presented in the structure generally accepted for the article. The discussion of the results is slightly expanded.
Reviewer 3 Report
The manuscript describes the synthesis of mixed carbides of the type Mo-W at temperatures of 700-800 ºC using sol-gel method. In this way high surface areas are obtained. The materials were extensively characterized. In my opinion the topic is of interest and can be accepted for publication after some minor revisions are made:
- The authors state several times that carbides are obtained at low temperature, I think this sould be revised since 700-800 ºC are typical values. Also, include some additional comparison with literature in the discussion section.
- In the inset graphs of Fig. 1, is it correct the X axes “Intensity distribution”?
- In Line 253 the authors state that “The absence of carbon established by results of XRD-analysis was also confirmed by SEM.” I do not understand this statement since it refers to metal carbides. Do authors refer to carbon deposits? If so, not sure if that could be rule out by SEM.
- What do authors mean by “Primary particles” are they aggregates?
- Line 293 “…carbon source in dispersions make it possible to formation of a carbides in one stage.” Should be …”the formation…”
Author Response
1. The manuscript describes the synthesis of mixed carbides of the type Mo-W at temperatures of 700-800 ºC using sol-gel method. In this way high surface areas are obtained. The materials were extensively characterized. In my opinion the topic is of interest and can be accepted for publication after some minor revisions are made:
1. The authors are grateful to the Reviewer for such a careful study of the manuscript.
2. The authors state several times that carbides are obtained at low temperature, I think this sould be revised since 700-800 ºC are typical values. Also, include some additional comparison with literature in the discussion section.
2. The authors agree with the comment of the Reviewer, the necessary corrections were made to the manuscript.
3. In the inset graphs of Fig. 1, is it correct the X axes “Intensity distribution”?
3. Axis labels are erroneous, necessary corrections have been made.
4. In Line 253 the authors state that “The absence of carbon established by results of XRD-analysis was also confirmed by SEM.” I do not understand this statement since it refers to metal carbides. Do authors refer to carbon deposits? If so, not sure if that could be rule out by SEM.
4. The authors mention free carbon, which is formed in the case of the synthesis of dispersions of molybdenum-tungsten blue with a large excess of the reducing agent. In this case, carbon is not present as deposits on the carbide surface, but is a separate phase - carbon particles with a smooth surface. The features of such Mo2C-C systems were discussed in detail in our previous article Gavrilova, N.; Dyakonov, V.; Myachina, M.; Nazarov, V.; Skudin, V. Synthesis of Mo2C by Thermal Decomposition of Molybdenum Blue Nanoparticles. Nanomaterials 2020, 10, 2053-2068.
5. What do authors mean by “Primary particles” are they aggregates?
5. Since the synthesis of carbides was carried out using dispersions of nanoparticles. It is logical to assume that the resulting material will be a corpuscular porous material. The micrographs of carbides show porous aggregates, which consist of primary carbides particles. Primary particles are formed from particles of molybdenum-tungsten blue, aggregates - when these primary particles are sintered.
6. Line 293 “…carbon source in dispersions make it possible to formation of a carbides in one stage.” Should be …”the formation…”
6. The authors agree with the comment of the Reviewer, the necessary corrections were made to the manuscript.
Round 2
Reviewer 2 Report
This report is not adequate for scientific writing and does not hold any scientific novelty and significance. Unfortunately, I could not give a positive comment in the current paper. So, I do not recommend the acceptance of this paper as a research paper.